# In Situ Ultraviolet Polymerization Using Upconversion Nanoparticles: Nanocomposite Structures Patterned by Near Infrared Light

**DOI:** 10.3390/nano10102054

**Published:** 2020-10-17

**Authors:** Hongsub Jee, Guanying Chen, Paras N. Prasad, Tymish Y. Ohulchanskyy, Jaehyeong Lee

**Affiliations:** 1College of Information and Communication Engineering, Sungkyunkwan University, Suwon 16419, Korea; hsjee@skku.edu; 2School of Chemistry and Chemical Engineering, Harbin Institute of Technology, Harbin 150001, China; chenguanying@hit.edu.cn; 3The Institute for Lasers, Photonics and Biophotonics and the Department of Chemistry, University at Buffalo, The State University of New York, New York, NY 14260, USA; pnprasad@buffalo.edu; 4College of Physics and Optoelectronic Engineering, Shenzhen University, Shenzhen 518060, China; 5School of Electronic and Electrical Engineering, Sungkyunkwan University, Suwon 16419, Korea

**Keywords:** upconversion, core-shell nanoparticles, photopatterning, ultraviolet, polymerization, near-infrared

## Abstract

In this paper, we report an approach to polymerization of a nanocomposite containing UV-polymerizable organic material and inorganic, NaYbF_4_:Tm^3+^ core-based nanoparticles (NPs), which are optimized for upconversion of near infrared (NIR) to ultraviolet (UV) and blue light. Our approach is compatible with numerous existing UV-polymerizable compositions and the NaYF_4_: Yb, Tm^3+^ core-based NPs are much more stable against harsh conditions than NIR organic photo-initiators proposed earlier. The use of a core-shell design for the NPs can provide a suitable method for binding with organic constituents of the nanocomposite, while maintaining efficient NIR-to-UV/blue conversion in the NaYbF_4_ core. The prepared photopolymerized transparent polymer nanocomposites display upconversion photoluminescence in UV, visible and NIR ranges. We also demonstrate a successful fabrication of polymerized nanocomposite structure with millimeter/submillimeter size uniformly patterned by 980 nm irradiation of inexpensive laser diode through a photomask.

## 1. Introduction

Photopolymerization is the light induced initiation of a chain polymerization process [1] and it is widely utilized in various fields, such as curing a coating film [2] and forming a planographic printing plate [3], as well as in applying to a resin letterpress printing plate [4], a printed circuit board [5] and in the field of the dentistry [6]. In most cases, photopolymerization is achieved by irradiation with ultraviolet (UV) or short-wave visible light [7], polymerization by red or near infrared (NIR) light has not been widely developed. In general, a photopolymerizable composition comprises a monomer and a UV absorbing photopolymerization initiator. The advantages of UV curing are its rapid polymerization controlled by light intensity [8], low energy consumption [9] and low temperature operation [10]. UV light cured polymers have high cross-linking density, they are very stable and have superior heat, chemical and mechanical resistance [11,12,13], and these characteristics allow for versatile applications. Use of a photomask enables a formulation to be polymerized in a well-defined geometrical area, producing a desirable pattern. The area exposed to the light is remained or removed, depending on that whether it is a positive or negative photoresist [14,15]. A proper development process involves removing the non-polymerized part of the formulation, thus allowing for producing a well-defined pattern of a photomasked polymerized material. Due to its advantages, UV curing is considered a promising method in the coating industry. However, because of strong absorption and scattering, UV radiation cannot penetrate deeply into the material structures; the degree of curing can be affected by the amount of transmitted light [16]. As a result, UV-induced photopolymerization is normally used for a thin-film system, and its applications in fabrication of thick layers and 3-dimensional (3D) is strictly hindered [17]. Chemical cross-linking can be further enhanced or modified by optimization of the resin formulations or adding reinforcing fillers such as alumina and silica mixing with polymers [18,19]. In recent times, polymer nanocomposites have emerged as promising hybrid organic-inorganic materials [20]. UV-curable polymer nanocomposites, based on the resin and inorganic NPs, have been studied to enhance the physical properties of UV-cured products [21]. This approach has been used to enhance many properties such as stiffness, thermal conductivity, electrical conductivity, barrier properties, refractive index and abrasion stiffness [22,23,24,25]. The optical transparency by the incorporation of NPs in polymer matrices is sometimes decreased after UV-curing [26]. Preventive strategies for agglomeration of the NPs in transparent polymer composite materials have been developed [27]; they include ligand exchange of the NPs’ stabilizing ligands as well as proper selection of polymeric hosts which inhibit agglomeration of the NPs during the polymerization. The decrease in transparency of the polymeric nanocomposite is especially manifested in the UV range which provide limited depth of the UV-induced photopolymerization of the polymeric nanocomposite materials. On the other hand, most used photopolymerization techniques involve high intensity UV light in the range of ~250–350 nm, which is known to be harmful to living cells and organisms [28,29], rendering them as hazardous to workers in manufacturing industries and making incompatible with many biomedical applications [30,31]. The development of photopolymerization approaches that utilize more biocompatible and deeper penetrating near infrared (NIR) light promises to overcome hurdles associated with UV polymerization. However, the reported NIR-absorbing photoinitiating systems are mostly based on cyanine dyes and their derivatives, remaining rather inefficient and rudimentary [32].

Another approach used to achieve NIR-sensitive photopolymerization is based on upconversion of NIR to UV light in lanthanide ions (e.g., Yb^3+^ and Tm^3+^) doped into glasses [31] or NPs [33,34,35]. Upconversion nanoparticles (UCNPs) based on a NaYF_4_:Yb^3+^,Tm^3+^ nanocrystalline core convert incident NIR light into UV or visible luminescence emissions, which, in turn, can trigger chemical reactions, including photolysis, photoisomerization, photo-coupling and photopolymerization. In UCNP-assisted polymerization, the upconverted UV/visible luminescence photons are reported to excite photosensitizers/photoinitiators that can subsequently initiate polymerization of monomers and crosslinkers [35].

Previously, we have demonstrated a facile approach to fabricate a new generation of UCNPs with enhanced upconversion luminescence in UV range. *α*-NaYbF_4_:0.5%Tm/CaF_2_ core/shell NPs were synthesized and these UV-emitting UCNPs were demonstrated to cause rapid in situ photochemical reactions in live cells under irradiation with a low-power NIR (975 nm) continuous wave (CW) laser diode [36]. The development of these UV enhanced UCNPs paved the way for accelerating the development of more efficient UV emitting UCNPs for the wide spectrum of applications, including UV polymerization. In this study, we propose to use the specially designed rare-earth ions containing core-shell NPs (NaYbF_4_:Tm^3+^ core-based) as nanofillers in a UV-sensitive material for curing and photopolymerization. These UCNPs are able to absorb NIR light from a CW light source in the range of ~900–1000 nm (peak at 970–980 nm) and upconvert it with high efficiency to UV light, which is able to produce photopolymerization in situ. The NaYbF_4_:0.5%Tm^3+^/NaYF_4_ core/shell UCNPs with extremely high efficiency of NIR-to-UV/blue upconversion were reported by us earlier, allowing us to apply them as NIR-to-UV/blue photon nanotransformers that enable intracellularly localized optogenetic activation of living cells with incident NIR light [37]. These UCNPs were now synthesized and introduced into a commercially UV curable material (photoresist SU-8 that contains UV photoinitiator of bisphenol A novolac glycidyl ether) followed by a successful NIR-induced polymerization. The resulting polymerized nanocomposite film is visually transparent and emits blue upconversion luminescence under NIR excitation; it was also found to be shaped according to the spatial distribution of the incident laser power density. We have also fabricated a polymerized nanocomposite geometric pattern of millimeter/submillimeter size via irradiation of a non-polymerized nanocomposite by 980 nm laser diode through a photomask.

## 2. Materials and Methods

The NaYbF_4_:0.5%Tm^3+^/NaYF_4_ core/shell UCNPs used in this work were synthesized as described previously [37], the synthesis involved the following three stages.

*Synthesis of α-NaYbF_4_: Tm^3+^ Core NPs*. The 1 mmol α-NaYbF_4_: Tm^3+^ core NPs were prepared via a thermal decomposition method at a high temperature. In a typical procedure, 0.5 mmol of RE_2_O_3_ (RE = Yb, Tm^3+^) were dissolved in 10 mL of 50% trifluoroacetic acid at 95 °C in a 250 mL flask for 1 h to yield a clear solution. Then the solution was evaporated to dryness under argon gas protection to obtain white powdered RE(TFA)_3_. Subsequently, 8 mL of oleic acid (OA), 8 mL of oleylamine (OM), 12 mL of 1-octadecene (ODE) and 2 mmol of sodium trifluoroacetate were added into the flask. The solution was heated to 120 °C and kept at that temperature for 45 min to remove water and oxygen, followed by heating to 300 °C at a rate of about 12 °C per min under argon gas protection and then kept at this temperature under vigorous stirring for 30 min. A syringe needle was used to let the argon gas out during the synthesis. Finally, the mixture was cooled to room temperature naturally, precipitated by excess ethanol, washed twice with a hexane/ethanol mixture, and collected by centrifugation at 6000 rpm for 5 min. The collected NPs were dispersed in hexane for further uses.

*Synthesis of β-NaYbF_4_: Tm^3+^ Core NPs.* Firstly, a total of 10 mL of OA, 10 mL of ODE, 2 mmol sodium trifluoroacetate and 1 mmol α-NaYbF_4_: Tm^3+^ core was added into a 250 mL flask. The resulting solution was then heated at 120 °C with magnetic stirring for 45 min to remove water and oxygen. The brown solution was then heated to 320 °C at a rate of about 12 °C per min under argon gas protection and kept at this temperature under vigorous stirring for 30 min. A syringe needle was used to let the argon gas out during the synthesis. Finally, the mixture was cooled to room temperature naturally, precipitated by excess ethanol and washed twice with a hexane/ethanol mixture, collected by centrifugation at 6000 rpm for 5 min. The collected 1 mmol β-NaYbF_4_: Tm^3+^ core NPs were dispersed in hexane for further uses.

*Synthesis of β-(NaYbF_4_:Tm^3+^)/NaYF_4_ Core-Shell NPs.* Firstly, a total of 0.5 mmol of Y_2_O_3_ was dissolved in 10 mL of 50% trifluoroacetic acid at 95 °C in a 250 mL flask for 1 h to yield a clear solution. Then the solution was evaporated to dryness under argon gas protection to obtain white powdered Y(TFA)_3_. Next, 10 mL of OA, 10 mL of ODE, 2 mmol sodium trifluoroacetate, and 1 mmol β-(NaYbF_4_: Tm^3+^) core was added into the flask. The resulting solution was then heated at 120 °C with magnetic stirring for 45 min to remove water and oxygen. The brown solution was then heated to 320 °C at a rate of about 12 °C per min under argon gas protection and kept at this temperature under vigorous stirring for 30 min. A syringe needle was used to let the argon gas out during the synthesis. Finally, the mixture was cooled to room temperature naturally, precipitated by excess ethanol, washed twice with a hexane/ethanol mixture and collected by centrifugation at 6000 rpm for 5 min. The collected NPs were dispersed in hexane for further uses. The resulted UCNPs with Yb/Tm ratio of 199:1 (core/shell NaYbF_4_: 0.5% Tm^3+^/NaYF_4_) were further utilized in this work.

*Photopolymerization.* To demonstrate the proof of principle for the NIR light-induced photopolymerization in situ, which occurs because of NIR-to-UV upconversion in the nanofillers, we have introduced the NaYbF_4_:Tm^3+^/NaYF_4_ core/shell NPs into a standard, UV-polymerizable formulation. The photoresist SU-8 (Miller Stephenson Chemical Co., Danbury, CT, USA) is a multifunctional, highly branched polymeric epoxy resin, which contains bisphenol A novolac glycidyl ether. UV-induced polymerization of this photoresist was used by us earlier [38,39]. A photoinitiator PC-2506 (Polyset Co., Mechanicville, NY, USA) undergoing a photochemical transformation upon absorption of a UV photon and generating a photoacid, was added to enhance cross-linking of the SU-8 using these NPs. To prepare the solution, the photoresist epon resin and a photoinitiator were added to the NPs suspension in cyclopentanone with a ratio of NPs 1: resin 5: photoinitiator 5: cyclopentanone 15. The overall mixture was stirred for 72 h and the prepared solution was coated on a glass substrate to make a 3~5 mm thick film. The coated sample was soft baked at 95 °C for 60 min to evaporate the solvent before laser exposure. After soft baking, the cooled down sample was irradiated with a 980 nm CW laser diode (Qphotonics, Ann Arbor, MI, USA) for 60 min at irradiation (power density of ~1 W/cm^2^) and the exposed sample was post baked for 30 min at 95 °C to accelerate cross-linking. A propylene glycol methyl ether acetate, (PGMEA, Sigma-Aldrich, St. Louis, MO, USA) was used for development, and sample was developed for 9 h.

For photopatterning, the photomask blank (Telic) comes with 120 nm thick chrome film and 530 nm thick A1518 photoresist was used and exposed to NIR light using the photomask. The exposed photoresist was developed with AZ351 developer (MicroChemicals, Ulm, Germany) followed by Cr-7 chrome etchant (Transene Co., Danvers, MA, USA) to transfer the pattern to the photomask blank.

*Photographic and confocal microscopy imaging.* Photographic images of the photopolymerized structures were taken by a digital camera (SX260 HS, Canon, Tokyo, Japan). The microscopy images of the photopatterned nanocomposite structure were obtained using a confocal fluorescence microscope (Leica TCS-SP2/AOBS) equipped with 980 nm CW laser diode. 10× air objective lens was employed and the laser power at the objective output was 11 mW, with a laser scanning frequency of 200 Hz. 

## 3. Results and Discussion

Figure 1 shows a concept of in situ UV induced photopolymerization using UCNPs. To make a patterned structure, a photomask was used to prevent unwanted crosslinking of the photoresist to the certain area by the light exposure. From the top, NIR light is incoming and is absorbed by UCNPs which upconvert NIR to UV light (a luminescence band peaked at ~350 nm) and the emitted UV light starts to crosslink the light exposed area.

We have increased rate of the NIR-to-UV upconversion by designing NaYbF_4_:0.5% Tm^3+^/NaYF_4_ core-shell NPs. Photoluminescence (PL) of these ~50 nm sized NPs under 980 nm excitation is mostly pronounced in the UV and the blue spectral regions which suggests extremely effective NIR-to-UV conversion and it should be very favorable for the UV-induced photopolymerization, as shown in Figure 2.

Figure 3a shows the photographic images of the polymer nanocomposite film that remained on the glass substrate after development when all non-crosslinked SU-8 was removed by the developer. As it can be seen in Figure 3a, a transparent film of the polymer nanocomposite was formed on the glass substrate, exhibiting visible to eye blue PL under excitation of 980 nm. It is important to know that the lens-like bulge in the center of the film is associated with higher power density in the center of irradiating laser beam (Gaussian profile of the beam intensity) [40]. A higher power density in the laser irradiation spot caused a brighter UV emission from the nanofillers and as a result, a thicker layer of the nanocomposite material could be polymerized. It should be also noted that the high efficiency of our UCNPs allowed us to use rather low doping concentration of NPs, which resulted in lesser light scattering and visual transparence of the polymerized nanocomposite.

We have further exploited the possibility to obtain a nanocomposite photopatterned with the NIR laser. Figure 3b shows a polymerized nanocomposite pattern obtained by 980 nm irradiation of the non-polymerized SU-8/UCNPs mixture through a photomask. As can be seen in the presented photo, an application of a photomask to the irradiated surface allowed us to obtain a geometric pattern of a polymerized nanocomposite, which displays blue upconversion emission under excitation with 980 nm laser diode. Figure 4 presents confocal microscopy images of the polymerized nanocomposite geometric pattern shown in Figure 3b, revealing structural details of the nanocomposite both in upconversion luminescence and transmission images.

It should be emphasized that the existing intense upconversion luminescence at 440~480 nm in addition to the UV emission (Figure 2b) can be also used for NIR light induced photopolymerization by upconverted light. This is especially important in the relevant applications, where 450 nm light sources are used for photopolymerization. For example, mixing the NPs with commercially available dental resin which is polymerizable with 450 nm light, the NIR curable nanocomposite can be obtained and it will allow to increase the depth of polymerization in comparison with the conventional blue-light curable resin. At the same time, doping of the inorganic NaYbF_4_:Tm^3+^ core-based NPs nanofillers in the dental resins can improve their mechanical properties. Furthermore, shells for the NaYbF_4_:Tm^3+^ core can be engineered to provide better binding strength with the organic constituents of the nanocomposite, while maintaining a high NIR-to-UV conversion efficiency. It is worth also noting that the blue emission of Tm^3+^-doped UCNPs (peaked at ~450 and ~475 nm) are overlapping with the absorbance of common commercial photoinitiators used in dental adhesive systems. At the same time, light at 980 nm penetrates deeper through dental ceramics compared to blue light, allowing for the use of blue emission from Tm^3+^ synthetized by Yb^3+^ ions under 980 nm to cure UCNP-doped resins in dental applications [41,42].

The advantage of our NaYbF_4_:Tm^3+^ core-based NPs is that these NPs are specially optimized for NIR-to-UV/blue conversion. This permits us to use a less powerful and inexpensive source of excitation which is normally operating at ~980 nm. The use of a core-shell design for the NPs allows to select and grow a shell on the core which can bind with the organic constituents of the nanocomposite, while maintaining efficient NIR-to-UV/blue conversion in the NaYbF_4_:0.5% Tm^3+^ core. Improved bonding between the resin matrix and inorganic NPs can be obtained because nanocomposites and their clusters have a much larger surface area per unit mass than conventional composites with microsized fillers, so having the nanosize of NPs is very important [43]. Furthermore, it is suggested that high surface area of the nanofiller affects the network forming kinetics of resin matrix [44].

## 4. Conclusions

The proposed photopolymerization approach utilizes NaYbF_4_:0.5% Tm^3+^ core-based NPs, which are optimized for NIR-to-UV/blue upconversion. Compared to organic dye-based NIR photoinitiators, NaYF_4_:Yb,Tm^3+^ core-based NPs are very stable due to their large anti-Stokes shift, photostability and single wavelength excitation. By mixing with the commercial photoresist (SU-8) that contains UV photoinitiator, upconversion NPs could be coated on a glass substrate and the coated mixture was polymerized by incident NIR laser diode light to produce polymerized nanocomposite in a bulk or a patterned structure. Because of lesser optical attenuation of NIR in comparison with visible light, small UCNPs’ size and their higher NIR-to-UV upconversion efficiency (which allows for lesser doping concentration) ~980 nm NIR light could penetrate from the top to the bottom of the resulted thick film of the polymerized nanocomposite and excite upconversion photoluminescence in the UV, visible and NIR ranges. Without or with the photomask, uniform polymerized structures were successfully fabricated with inexpensive laser diode operating at ~980 nm. This research extended the usage of polymerized film-based applications, since the harmful UV light source could be replaced by inexpensive and NIR light sources, and paved the way to NIR-induced fabrication of polymerized nanocomposite structures in the emerging applications.

## Figures and Tables

**Figure 1 nanomaterials-10-02054-f001:**
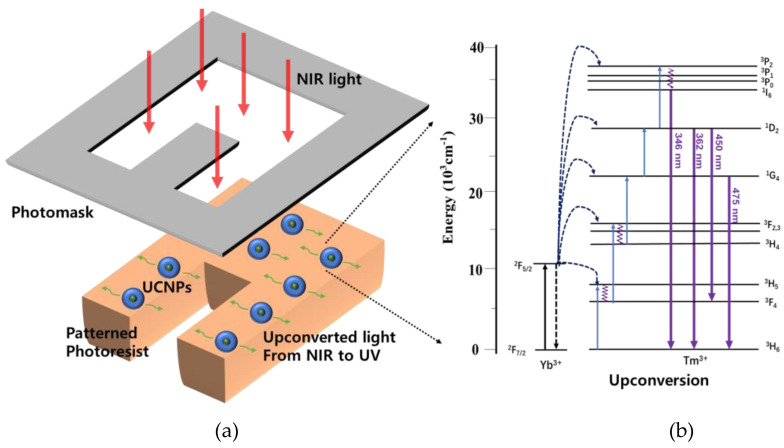
(**a**) Concept of a near infrared (NIR)-induced photopatterning of Upconversion nanoparticles (UCNPs)-doped polymerizable nanocomposite, (**b**) Simplified scheme illustrating energy levels of Tm^3+^ dopant ions and the corresponding transitions. ^1^D_2_–^3^F_4_.

**Figure 2 nanomaterials-10-02054-f002:**
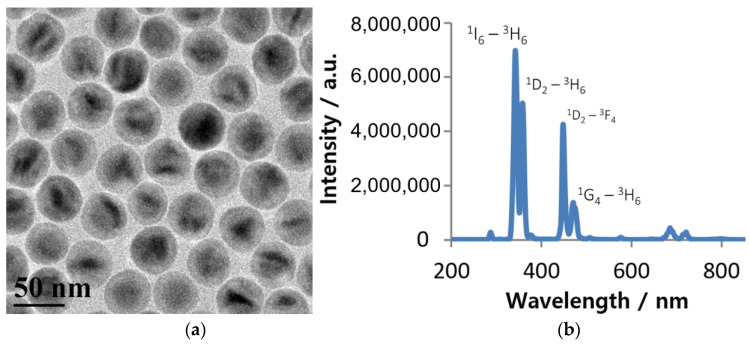
(**a**) Transmission electron microscopic (TEM) image of hexagonal NaYbF_4_:Tm^3+^/NaYF_4_ NPs, displaying the clear core-shell structure, (**b**) Upconversion luminescence of colloidal hexagonal NaYbF_4_:Tm^3+^/NaYF_4_ NPs dispersed in hexane, showing intense UV and blue emission under 980 nm excitation, along with the corresponding transitions in Tm^3+^ ions (Figure 1b).

**Figure 3 nanomaterials-10-02054-f003:**
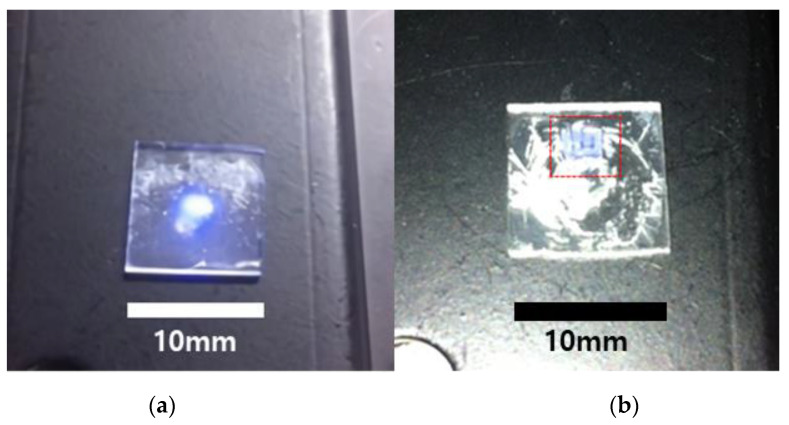
Photographic images of the nanocomposite film (**a**) and the geometric pattern (**b**) of the SU-8 based nanocomposite polymerized under the 980 nm excitation (marked by the dotted red line). The visible blue color is from the upconversion luminescence of the nanocomposite excited by the laser diode at 980 nm.

**Figure 4 nanomaterials-10-02054-f004:**
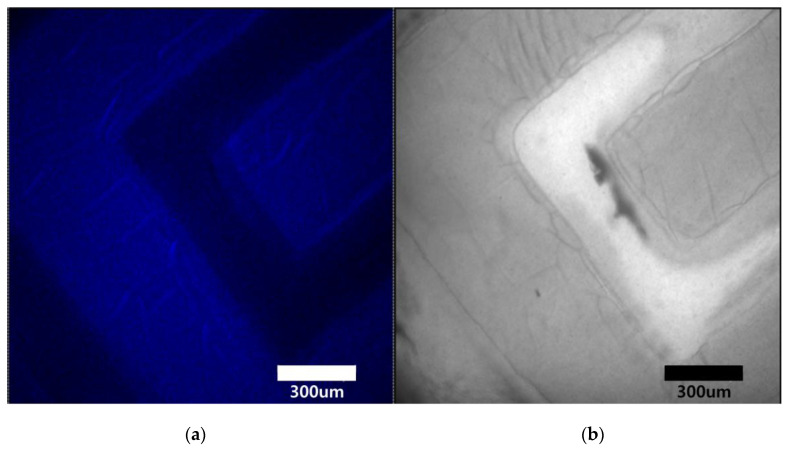
Confocal microscopy images of the NIR patterned polymer nanocomposite. Upconversion photoluminescence (PL) (**a**,**b**) transmission images are shown.

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
