# Peer review of "In Situ Ultraviolet Polymerization Using Upconversion Nanoparticles: Nanocomposite Structures Patterned by Near Infrared Light"

_nanomaterials, 2020, doi:10.3390/nano10102054_

Round 1

Reviewer 1 Report

To my opinion, this paper can be published in Nanomaterials after minor revision. Here are the comments to take into account for the revised version:

  1. The numbers of the chemical formulae must be written in superscript or subscript.
  2. What is the molar ratio between Yb3+ and Tm3+?
  3. The authors must add the XRD patterns evidencing the formation of a and β crystalline phases for NaYbF4 cores. By the way, what is the interest in preparing a-NaYbF4 since theses cores are not used for core-shell synthesis? In addition, the authors must justify the choice of β-NaYbF4 as cores for the core-shell systems.
  4. All the acronyms (OA, OM and ODE) must be defined at least once in the text.
  5. In Figure 2, it would be relevant to add the nature of theTm3+ transitions on the emission spectrum.
  6. Figure 3(b) is unclear.
  7. why did you write constituent in capital letters line 188?
  8. Check the chemical formula line 189.
  9. Line 189, check once again the chemical formula, Y atoms have been used instead of Yb ones.

Reviewer 2 Report

In this manuscript, authors used NaYbF4: Tm core-based NPs which optimized to NIR-to-UV/blue upconversion for the photopolymerization. It had photostability and single wavelength excitation. Author announced Photopatterned structures for biocompatible applications. But there is no any bio application report in this paper. It is a main concern of this paper. However, it is well-written and has good structure. It can be published after major revise.

Reviewer 3 Report

The Authors report on the formulation of NaYB4:Tm core-shell NPs for light frequency upconversion from NIR to UV and increase de penetration depth of the radiation.

The topic is interesting and well motivated, I suggest to improve the structure and clarity of the text to facilitate the reading and improve the impact of the work.

Figures should be also improved as follows:

In figure 1 please explain better what the two levels concerned by NIR transitions.

In figure 2b name the axis

In figure 3b the photo is not clear

In figure 4 the caption should be more accurate

Moreover authors speak about application in dentistery but they don't give any experimental justification for this, if this is a possible future application, this should go in the conclusions section.

Round 2

Reviewer 2 Report

It can be published.

Reviewer 3 Report

The revisions provided in the revised MS are sufficient for answering my main concerns, I suggest the publication of the MS in its present form.

Regards